# Management of HPV-Related Squamous Cell Carcinoma of the Head and Neck: Pitfalls and Caveat

**DOI:** 10.3390/cancers12040975

**Published:** 2020-04-15

**Authors:** Francesco Perri, Francesco Longo, Francesco Caponigro, Fabio Sandomenico, Agostino Guida, Giuseppina Della Vittoria Scarpati, Alessandro Ottaiano, Paolo Muto, Franco Ionna

**Affiliations:** 1Head and Neck Medical Oncology Unit, Istituto Nazionale Tumori, IRCCS G. Pascale, 80131 Naples, Italy; f.caponigro@istitutotumori.na.it; 2Division of Surgical Oncology Maxillo-Facial Unit, Istituto Nazionale Tumori—IRCCS—Fondazione G. Pascale, Via Mariano Semmola, 80131 Naples, Italy; f.longo@istitutotumori.na.it (F.L.); a.guida@istitutotumori.na.it (A.G.); f.ionna@istitutotumori.na.it (F.I.); 3Unit, Istituto Nazionale Tumori—IRCCS—G. Pascale, 80131 Naples, Italy; f.sandomenico@istitutotumori.na.it; 4Medical Oncology Unit, Ospedale Pollena, ASL NA3, 80040 Naples, Italy; giuseppina.dellavittoria@gmail.com; 5SSD Innovative Therapies for Abdominal Metastases, Department of Abdominal Oncology, INT IRCCS Fondazione G. Pascale, 80131 Naples, Italy; a.ottaiano@istitutotumori.na.it; 6Radiation Oncology Unit, Istituto Nazionale Tumori—IRCCS—G. Pascale, 80131 Naples, Italy; p.muto@istitutotumori.na.it

**Keywords:** human papilloma virus, squamous cell carcinoma of the head and neck, translational research, oncogenes, tumor suppressor genes

## Abstract

Head and neck squamous cell carcinomas (HNSCCs) are a very heterogeneous group of malignancies arising from the upper aerodigestive tract. They show different clinical behaviors depending on their origin site and genetics. Several data support the existence of at least two genetically different types of HNSCC, one virus-related and the other alcohol and/or tobacco and oral trauma-related, which show both clinical and biological opposite features. In fact, human papillomavirus (HPV)-related HNSCCs, which are mainly located in the oropharynx, are characterized by better prognosis and response to therapies when compared to HPV-negative HNSCCs. Interestingly, virus-related HNSCC has shown a better response to conservative (nonsurgical) treatments and immunotherapy, opening questions about the possibility to perform a pretherapy assessment which could totally guide the treatment strategy. In this review, we summarize molecular differences and similarities between HPV-positive and HPV-negative HNSCC, highlighting their impact on clinical behavior and on therapeutic strategies.

## 1. Background

Head and neck squamous cell carcinoma (HNSCC) is an heterogeneous group of malignancies comprised of several entities arising from different anatomical subsites, including the oral cavity, hypopharynx, oropharynx, larynx and nasopharynx. They are characterized by different etiologies, genetics and clinical behaviors. HNSCC development is strongly related to tobacco and/or alcohol consumption and/or oral trauma, and/or, in particular in some countries such as Iran and Southeast Asia, betel quid chewing. Nevertheless, in the last 15 years, remarkable changes in HNSCC epidemiology have been observed and a critical increase in the diagnosis of some kinds of HSNCC, e.g., the oropharyngeal carcinoma, have been noted. This feature is probably due to the increasing incidence of human papilloma virus (HPV)-related tumors [1,2,3]. Several lines of evidence support the existence of at least two genetically different types of HNSCC, one virus-related and the other alcohol and/or tobacco and oral trauma-related, characterized by both clinical and biological opposite features [3,4]. Unlike HPV-negative HNSCC, HPV-positive HNSCC often occurs in younger patients with minimal or no tobacco exposure [5,6]. HPV-positive HNSCC, similarly to its HPV-negative counterpart, has a male predominance, with men suffering a three-to-five times higher incidence than women worldwide [7].

HPV-positive HNSCC carries a favorable prognosis if compared to HPV-negative tumors. In fact, five-year survival rates for patients with advanced-stage HPV-positive HNSCC are 75–80%, versus values less than 50% in patients with similarly staged HPV-negative tumors [8,9]. The cause of the aforementioned different behavior is the different chemo- and radiosensitivity shown by the HPV-positive and HPV-negative HNSCCs. In fact, several clinical trials have shown that HPV-positive HNSCC patients have a better response to chemotherapy and radiation therapy than HPV-negative cases [10,11,12]. The reasons for this different behavior should be searched in the opposite genetic features which characterize the two types of tumors.

In this review, we will analyze the genetics of both HPV-positive and HPV-negative HNSCC, highlighting their impact on the clinical behavior and finally on the therapeutic strategies.

## 2. Genetics of HPV-Positive HNSCC

Carcinogenesis, which is the complex process through which the normal cell is pushed to transform itself into a cancer cell, is very different between HPV-related and non-HPV-related HNSCC. Viral carcinogenesis in HNSCC is partly due to HPV infection, with the oropharynx being the most commonly involved site. HPV-mediated carcinogenesis is driven by a few viral oncoproteins expressed by high-risk HPV genotypes. In particular, E6 and E7 have shown to inactivate p53 and retinoblastoma protein (pRb), respectively, and to impair several metabolic pathways in the infected cells [13,14,15].

The dominant viral type associated with the development of HNSCC (especially oropharyngeal carcinoma) is HPV16, while HPV18, 31 and 33 are less frequently detected [16,17]. The commonly acknowledged paradigm of HPV carcinogenesis, based on studies conducted upon uterine cervical cancer, highlights the importance of HPV genome integration as a premalignant lesion [18]. Nevertheless, recent acquisitions support the issue that nearly 30% of HPV-positive HNSCC contained only episomal HPV, prompting new theories about the alternative mechanisms of HPV-driven carcinogenesis.

The HPV life cycle is related to the host cell capability of proliferation, since its genome does not encode the polymerase or other enzymes necessary for viral replication; so HPV, once entered in the cell, both by integrating itself in the host DNA remaining in the form of episomes, becomes able to promote cell cycle progression mainly through the above mentioned genes, E6 and E7 [19,20]. In particular, E6 inactivates p53, affecting its capability to activate p21, which, in turns, blocks CDK (Kinase Cyclin Dependent)/Cyclin heterodimers. This latter protein, through phosphorylation of pRB, provokes E2F release. E2F is able to act as an oncogene stimulating the G1/S transition. The final result is a deregulation of the cell cycle [21,22]. On the other hand, E7 allows for retinoblastoma protein (pRB) degradation, which in normal conditions binds and puts off the transcriptional factor E2F. Moreover, pRB degradation is able to remove the inhibitory feedback upon p16 synthesis, thus leading to its hyperproduction. For this reason, in HPV-infected tumor cells, p16 is always upregulated [23,24]. The cell cycle dysregulation caused by HPV infection can lead to accumulation of DNA damage, and thus promotes carcinogenesis (see Figure 1).

The E7 protein is also able to induce methylation of suppressors with a morphogenetic effect on genitalia (SMG-1) gene promoter, provoking its dysfunction. SMG-1 is a tumor suppressor gene encoding for a protein able to arrest the cell cycle in response to DNA damage [25,26].

In addition, HPV DNA often integrates in host DNA, and the integration takes place in specific loci, such as RAD51. This nonrandom integration leads to RAD51 gene dysfunction. RAD51 is an enzyme involved in double-strand DNA break repair, and its function results in impairment in virus-related HNSCC [27].

Overall, from a genetic point of view, HPV-related tumors present unique features, such as p16 overexpression, CyclinD1 and pRB down-regulation, a low EGFR (Epithelial growth factor receptor) expression with a high proliferating index (Ki-67). Interestingly, the most mutated and disrupted pathway in those tumors is the Akt-related one [28,29].

HPV-related carcinogenesis is characterized by a relatively low number of DNA mutations and chromosomal changes while, on the other hand, by a higher percentage of epigenetic changes. As a matter of fact, virus-related HNSCCs are associated with a significantly lower mutational rate, when compared to non-virus-related HNSCC [30,31,32]. The lower tumor mutational burden (TMB) typical of HPV-related HNSCC leads to a generation of oligoclonal tumors which are intrinsically more chemo- and radiosensitive. Table 1 describes the main genetic differences between HPV-positive and the HPV-negative HSNCC.

On the other hand, in the HPV-negative (mutagen-related) HSNCCs, the gradual acquisition of mutations involving both “oncogenes” and “tumor suppressor genes”, by effect of the mutagens contained in alcohol, tobacco and betel, is critical to cause neoplastic transformation and is reflected by their high mutation load. Thus, the pathogenesis of mutagen-related HSNCC is strongly linked to progressive accumulation of mutations which affect several DNA traits. Interestingly, these DNA changes often concern some important and crucial genes, namely TP53, CCND1, INK-4, EGFR, and NOTCH1. Mutagen-related HSNCC often presents EGFR and CCND1 amplification, INK-4 mutations, inducing p16 down-regulation, and disruption of the NOTCH-1 pathway [4]. The wide number of genic aberrations generates heterogeneous neoplastic populations, which are responsible for chemo- and radioresistance often observed in alcohol- and tobacco-related SCCHN.

In addition, it is very important to discern between HPV-positive and HPV-“related” HNSCC. These two categories of tumors may be very different from each other, due to the fact that in the HPV-related HNSCC, the entire carcinogenesis process is initiated and sustained by HPV, which may be defined as the sole “driver”. In this case only, the tumor is characterized by a low TMB, a wild-type p53 status, p16 overexpression and the concomitant wild-type status of the genes INK4 (which encodes for p16), EGFR and CCND1 (the gene encoding for cyclin D1). Nevertheless, there is another category of HPV-positive tumors which harbor p53 mutations, CCND1 amplification and INK4 mutations, too. The latter have a poor prognosis, similarly to the HPV-negative HNSCC.

Weinberger et al. [33] performed a molecular analysis on 80 cases of oropharyngeal carcinomas, analyzing the status of a panel of biomarkers, such as p16, p53 and pRB and correlating it with the clinical outcome. On the basis of the results obtained, they divided HNSCCs into three classes. Class I was characterized by the absence of the HPV, and the contemporaneous presence of p53 mutations and p16 inactivation; these tumors were considered to be mutagen-related and showed poor prognosis. Class II was characterized by HPV positivity, and in concomitance, p53 mutations and p16 inactivation; these were considered to be HPV-positive but not HPV-related, and showed poor prognosis. Finally, class III encompassed all the HPV-related HNSCCs characterized by HPV positivity, wild-type for p16 and p53. Figure 2 describes the classification of oropharyngeal carcinomas suggested by Weinberger et al.

This last classification may be of help in improving the identification of the virus-related HNSCCs and in separating them from their mutagen-related counterpart, aiming to modulate the therapy options.

## 3. HPV-Related Tumors: Implications in Clinical Practice

Due to the high chemo- and radiosensitivity shown by HPV-related HNSCC, some authors have hypothesized the possibility of under-treating affected patients. The rationale for de-intensification of chemoradiotherapy is to reduce the side-effects caused by combination therapy. In fact, radiotherapy is associated with dose-related adverse side effects, from acute toxicities like mucositis and loss of taste to long-term problems including renal dysfunction, severe dysphagia, significant xerostomia, hearing loss, osteoradionecrosis, strong neck muscle fibrosis, accelerated arteriosclerosis and trismus. These toxicities may cause a cascade of events, such as infections, dysphagia, feeding tube necessity and increased hospitalizations, that can markedly affect the quality of life. All the above side effects are strongly boosted up by the addition of chemotherapy. Thus, it is deductive to think that the reduction of the dose of radiation therapy and/or the chemotherapy may reduce the percent of cumulative toxicities.

In 2014, Cmelak et al. presented at ASCO (American Society of clinical Oncology) the preliminary results of phase II trials enrolling HPV-positive HNSCC patients. Patients underwent induction chemotherapy followed by two different regimens of concurrent cetuximab radiotherapy (RT) on the basis of the obtained response. In particular, patients who completely responded to induction chemotherapy were treated with an underpowered RT regimen, consisting in 50 Gy instead of the standard 70 Gy, while concurrent cetuximab and 70 Gy RT was administered in those who showed only a partial response. As a result, a better outcome was obtained by patients treated with underpowered RT [34]. This study paved the way to different trials assessing the de-intensification strategies in HPV-related HNSCC.

Chera BS et al., in a prospective phase II trial, enrolled 44 patients with diagnosis of T0-T3, N0-N2c, M0, p16-positive HNSCC and treated them with 60 Gy of intensity-modulated radiotherapy with concurrent weekly intravenous cisplatin. As a result, 3-year local control, regional control, cause-specific survival, distant metastasis-free survival, and overall survival rates were 100%, 100%, 100%, 100%, and 95%, respectively. The authors concluded that, in HPV-HNSCC patients, a protocol consisting of under-dosed radiation therapy (IMRT) given concomitantly with weekly cisplatin was able to obtain a good preservation of quality of life and an excellent 3-year tumor control and survival [35].

Woody NM et al. treated a cohort of patients with HPV-positive oropharyngeal squamous cell carcinoma with definitive chemoradiotherapy (70–74.4 Gy) to the primary site and, since a postradiation neck dissection was planned, 54 Gy to the involved nodal areas. The authors observed a five-year locoregional control, disease-free survival and overall survival of 96%, 81% and 86%, respectively. The conclusion was that regional lymph node control in HPV-positive oropharyngeal cancer was not compromised by a de-escalated dose of radiotherapy to involved nodes in the setting of concurrent cisplatin-based chemotherapy [36].

These aforementioned trials, although being considered positive due to the good activity shown by the de-intensified treatments, suffered from being only phase II and nonrandomized trials.

Onita et al. [37] performed a retrospective review of patients with p16-positive oropharyngeal carcinomas who underwent, from 2006 to 2016, definitive radiotherapy concurrently with either triweekly cisplatin (*n*  =  251) or cetuximab (*n*  =  40). The study comprised also patients with stage I disease. Median follow-up was 40 months. On multivariate analysis comparing cisplatin and cetuximab, the 3-year locoregional recurrence (LRR) was 6% vs. 16% (*p* = 0.07); the 3-year distant metastasis rate (DM) was 8% vs. 21% (*p*  =  0.04), the 3-year overall recurrence rate (ORR) was 11% vs. 29% (*p* = 0.01), and the 3-year cause-specific survival (CSS) was 94% vs. 79% (*p* = 0.06), respectively. The aforementioned results sharply favored the cisplatin arm. Nevertheless, when a stage-based subgroup analysis was done, the results were interesting; in fact, for stage I-II patients, 3-year LRR, DM, ORR and CSS did not significantly differ. The same parameters were significantly superior in the cisplatin arm, only when the authors considered stage III diseases. The authors concluded that, when given concurrently with radiotherapy, cetuximab and triweekly cisplatin demonstrated comparable efficacy for stage I–II p16-positive oropharyngeal squamous carcinomas (OPSCC). However, cetuximab appeared to be associated with higher rates of treatment failure and cancer-related deaths in stage III disease. Lately, Gillison ML et al. [38] published the results of a large (987 patients enrolled) prospective phase III trial comparing concurrent cisplatin (at the standard dose of 100 mg for square meter of body surface) and 70 Gy radiation therapy, with the combination of cetuximab and the same radiation therapy regimen. This trial aimed to demonstrate the noninferiority of cetuximab and radiotherapy with respect to the standard of care, namely cisplatin-radiation therapy, in a population of patients affected by locally advanced HPV-related HNSCC. As a result, the experimental combination of cetuximab and radiation therapy did not meet the primary endpoint, showing to be inferior to the standard cisplatin radiotherapy (estimated 5-year overall survival was 77.9% in the cetuximab group versus 84.6% in the cisplatin group). Similar results were obtained also with regard to progression-free survival. The authors concluded that for patients with HPV-positive oropharyngeal carcinoma, radiotherapy plus cetuximab showed inferior overall and progression-free survival if compared to radiotherapy plus cisplatin, so that in those patients, radiotherapy plus cisplatin remained the standard of care (for eligible patients).

Mehanna H et al. performed a very similar trial aiming to demonstrate the noninferiority of the cetuximab-radiotherapy combination, in comparison with cisplatin and radiotherapy (De-ESCALaTE HPV trial). They enrolled 334 patients with locally advanced HPV-positive oropharyngeal carcinoma and randomized them to receive standard 70 Gy radiation therapy associated with cetuximab or cisplatin (100 mg for square meter of body surface). The results were similar to the previously mentioned trial, showing the significant less efficacy of cetuximab if compared with the standard cisplatin (2-year overall survival 97.5% in the cisplatin arm vs. 89.4% in the cetuximab arm, *p* = 0.001) [39].

Recently, Jones et al. published the updated results of the aforementioned phase III De-ESCALaTE HPV trial. Three hundred and thirty-four (334) patients were randomized to cisplatin (166) or cetuximab (168). Two-year overall survival (97.5% vs. 90.0%, HR: 3.268, *p* = 0·0251) and recurrence rates (6.4% vs. 16.0%, HR: 2.67; *p* = 0.0024) favored the cisplatin arm. Furthermore, the results of this phase III large trial highlighted that in HPV-positive patients, the standard association of cisplatin-radiotherapy should not be avoided [40].

On the basis of the conflicting results obtained in clinical trials, de-intensification therapies have not been taken into account in clinical practice, and moreover, it is not yet clear if the de-intensification should involve systemic therapy or radiation therapy. Table 2 shows the main studies exploring the concept of de-intensification of the standard chemoradiotherapy regimen in patients with HPV-positive HSNCC.

Overall, HPV status has a prognostic significance in HSNCC, but it has not yet altered the treatment guidelines. As a matter of fact, the last version of the National Comprehensive Cancer Network (NccN) Guidelines has sharply separated treatment pathways for p16-positive and p16-negative oropharyngeal carcinomas, but the treatment options for p16-positive and p16-negative oropharyngeal carcinomas are almost identical, with the below-mentioned differences only.

HPV-positive oropharyngeal carcinomas staged as T1 N1 M0, which in the previous version of the guidelines were suitable for chemoradiotherapy, should be treated now with upfront surgery or alternatively with radiation alone. T2 N1 M0 tumors (with a single <3 cm lymph node metastasis) may be suitable for chemoradiation, but concomitant chemoradiation is considered to be only a 2B (namely, not strongly supported by evidence) category of choice [41]. The surgical approach that should be chosen is the trans oral robotic surgery (TORS) which in clinical trials is able to guarantee the same efficacy at a price of a significantly minor morbidity [42,43].

Another difference between HPV-positive and HPV-negative oropharyngeal carcinoma takes into account the role of adjuvant chemoradiotherapy for the so-called “high-risk” disease. Adjuvant concurrent chemoradiation represents a category 1 recommendation for all the patients surgically resected and with presence of extranodal extension. Nevertheless, recent data have highlighted that HPV-related carcinomas staged as T1–2 N1 (single <3 cm metastases) M0, should undergo adjuvant radiotherapy alone [44,45]. On these bases, the NCCN panel of experts recommend the omission of chemotherapy in concomitance with adjuvant radiotherapy in patients with T1–2 N1 M0 HPV-related oropharyngeal carcinoma surgically resected with extranodal extension; nevertheless, this recommendation is only a 2B category option.

## 4. HPV Infection and Immunotherapy

Immunotherapy is a therapeutic strategy aiming to reinforce the host immune system, helping it in reacting against tumor cells. The entire rationale of the immunotherapy has its roots in the existence of the so-called “tumor-associated antigens” (TAAs), namely protein antigens exposed by the tumor cells, able to elicit a strong immune response. Several strategies of immunotherapy are available in clinical practice and others are still being tested, but the most recent immunotherapeutic drugs are those acting in the “checkpoint” phases. The two well-acknowledged checkpoint phases are the “priming phase” (during which the naïve T-lymphocytes mature and became able to attack the tumor cells) and the “effector phase” (during which the matured T-lymphocytes attack and destroy the tumor cells, by recognizing the TAA).

Virus-related and mutagen-related HNSCCs display different genetic and immunologic features. Some data indicate that tobacco and alcohol, provoking several DNA mutations, also alter the tumor immune microenvironment. These last immune microenvironment alterations strongly affect tumor response to immunotherapy, thus leading to lower response rates after immunotherapy [46]. Desrichard et al. demonstrated a significant correlation between a specific “smoking-signature”, which characterizes the smoke-related HNSCC, and the entity of the tumor-immune-infiltrate. In particular, they observed that a specific smoke-associated signature (the signature defined by Alexandrov), characterized by a wide number of DNA changes and a very high mutational burden, significantly correlated to a low immune tumor infiltrate. Interestingly, this signature also related to poor response to immunotherapy [47,48]. On the other hand, the HPV-related HNSCC subgroup showed the opposite features, being characterized by a robust CD8 lymphocyte-mediated response and a better response to immunotherapy. The main implications of the aforementioned features are that the HPV-related HNSCC responds better to immunotherapy, and in particular, to checkpoint inhibitors, while the mutagen-related one does not, being characterized by a noninflamed phenotype.

## 5. Future Implications on Therapy

The future therapeutic approaches should start from the statement that HPV-related HNSCC represents an entity which is very different genetically to mutagen-related HNSCC, mainly due to their intrinsic chemo- and radiosensitivity. Consequently, when, in clinical practice, there is a doubt whether to choose surgery or chemoradiotherapy, we can hypothesize, to address HPV-related tumors, that conservative treatment instead of surgery be used, especially if the surgical procedure is burdened with greater compromise of quality of life. On these bases, our effort should be aimed at rapid identification of the HPV-related tumors, in particular those belonging to the class III, as reported by Weinberger et al. [33]. According to more and more data [49,50,51], the latter seem to be characterized by a genetic signature, namely a pattern of genetic and epigenetic changes, which may sharply distinguish them from the other non-virus-related tumors. HPV-related HNSCC (in particular, oropharyngeal cancers) often shows the following features—P16 overexpression, low EGFR expression, wild-type TP53 and low CyclinD1 expression; in addition they show a lower TMB and an higher number of epigenetic changes, if compared with the mutagen-related counterpart [4,52,53].

Further complicating the matter is the molecular heterogeneity existing within HPV-positive tumors. As a matter of fact, a 2014 study of HNSCC from The Cancer Genome Atlas (TCGA) found that in a group of 35 HPV-positive tumors, 25 had integration of the viral genome while 10 tumors lacked integration [54] Further data have highlighted that nearly 30% of HPV-positive oropharyngeal carcinomas contained only episomal HPV [55]. Oropharyngeal cancers showing integrated versus nonintegrated HPV have differences in somatic gene methylation, gene expression patterns, mRNA processing, and inter- and intrachromosomal rearrangements [56]. In a recent biomolecular analysis of a subgroup of HPV-positive HNSCC, authors identified the presence of deletions or mutations of two proteins that inhibit NF-kB and activate interferon, TNF receptor-associated factor 3 (TRAF3) and cylindromatosis (CYLD) [57]. Furthermore, the presence of these DNA changes was related to the prognosis, with survival improved for patients whose tumors carried defects in either TRAF3 or CYLD. Conversely, the survival of HPV-positive patients without these mutations was similar to that of HPV-negative patients [58].

TRAF3 and CYLD gene deletions or disruptive mutations were identified in 28% of HPV-positive specimens in the initial TCGA HNSCC cohort and it correlated to the absence of HPV gene integration and decreased tobacco exposure [59], leading to the consideration that both DNA damage and the presence of reactive oxygen species (induced by tobacco mutagens) may favor HPV integration.

In conclusion, we can assert that the positivity for HPV (p16 test) is not enough to consider the tumor as HPV-related, and other markers should be taken into account for this scope.

A subgroup analysis carried out in the TAX 324 study, as well as the results of the ECOG 2399 trial, clearly demonstrated that HPV-related HSNCC responded better to induction TPF (docetaxel, cisplatin and 5-FU) followed by chemoradiation, when compared with the p16-negative counterpart [60,61], (Figure 3).

Moreover, Feng et al. [62] demonstrated that wild-type (WT) CCND1 (the gene encoding for CyclinD1) HNSCC displayed a significantly better response to induction chemotherapy compared with tumors showing CCND1 gene amplification (Figure 4). HPV-related HNSCCs often show the wild-type status for CCND1 concomitantly with wild-type status of p16. The authors concluded that the WT status for CCND1 could predict good response to induction chemotherapy and, consequently, we can assume that the presence of HPV-related carcinogenesis may represent a predictive factor of good response to induction chemotherapy, followed by chemoradiation (Figure 4).

The two latter studies have furtherly highlighted the concept that HPV-related tumors not only have a better prognosis but also a better response to conservative treatments when compared with the HPV-negative counterpart.

HPV presence in the tumor cells could also guide the immunotherapy strategies. Starting from the hypothesis that viral antigens are much more immunogenic than those “self”, a number of clinical trials have tested vaccination strategies which selectively target the viral antigens, such as E6 and E7 proteins [63]. Results are encouraging but data are still immature.

Regarding the use of the checkpoint inhibitors, only nivolumab and pembrolizumab are presently approved drugs for the treatment of recurrent/metastatic HSNCC. Both the drugs are indifferently employed in HPV-positive and HPV-negative patients. Nevertheless, it is interesting to highlight that a subgroup analysis in the context of the Keynote 141 trial (the study that led to the approval of nivolumab in clinical practice) has revealed that, among patients with HPV-positive tumors, the median OS was 9.1 months for patients treated with nivolumab versus 4.4 months for those treated with the standard-therapy, confirming the possibility that virus-related HSNCC better responds to checkpoint inhibitors [64]. More data are needed to assert the aforementioned issue, but it seems clear enough that virus-related HSNCC is more suitable to respond to immunotherapy compared with its mutagen-related counterpart.

## 6. Conclusions

HNSCCs are a very heterogeneous group of tumors affecting more than 65,000 patients per year in the United States. Mortality is strongly related to the initial staging, with both advanced and locally-advanced diseases having a poor prognosis. Lately, the knowledge of HNSCC genetics, as well as the translational research in this field, have gained more and more importance in the management of patients. Thus, the discovery of a subgroup of HNSCC, HPV-related HNSCC, particularly different from the others, paved the way to a different approach to HNSCC in clinical practice.

As largely demonstrated by scientific literature, HPV-related tumors are much more radiosensitive and chemosensitive when compared with their mutagen-related counterpart, and this feature can significantly impact on the clinical management of the patients.

Nevertheless, there are at least two problems to face—the importance of sharply distinguishing the HPV-related tumors from the non-HPV-related ones, independently from the presence of the viral DNA in the tumor cells, and the possibility to employ this information in clinical practice.

The identification of the viral protein E6 and E7 may be the best way to identify virus-related HNSCC [15], but this methodology does not take into account some important considerations. As a matter of fact, there is a subgroup of HPV-positive HNSCC that displays both viral DNA and E6/E7 proteins which is characterized by nonviral carcinogenesis. In this last case, some particular DNA and chromosomal changes such as p53, CCND1 and EGFR mutations, as well as a high TMB, are often present. In these cases, the carcinogenesis is due to mutagens from alcohol and tobacco, and the presence of HPV is not relevant, with these tumors having a prognosis comparable with those that are HPV-negative.

Different markers, other than p16, have been taken into account with the aim to best identify the HPV-driven carcinogenesis—TP53, pRB and CCND1, with their expression being very peculiar in HPV-related HNSCC (class III according to Weinberger).

The second and most relevant problem to be solved is the applicability of the aforementioned information in clinical practice. In fact, the last TNM version distinguishes between HPV-related (p16-positive) and non-HPV-related tumors, highlighting the impact that HPV has on the prognosis. Nevertheless, the therapeutic strategies used for HPV-related oropharyngeal cancers are almost the same as in non-HPV-related tumors, with few exceptions.

According to the latest translational acquisitions, we can speculate that in the near future, the HPV-related HNSCC could have different treatments when compared with the mutagen-related tumors. In particular, the locally advanced virus-related HNSCC could be treated with conservative strategies in spite of radical surgery, being very chemo and radiosensitive. On the same bases, HPV-related tumors, which are more suitable to respond to immunotherapy, could benefit from a single drug immunotherapy, such as checkpoint inhibitors; the mutagen-related counterpart, which often have a noninflamed phenotype, necessitates stronger immune modulation with two or more immunotherapeutic drugs.

Further studies on translational research should be designed with the aim to discern therapeutic options on the bases of the genetics of tumors.

## Figures and Tables

**Figure 1 cancers-12-00975-f001:**
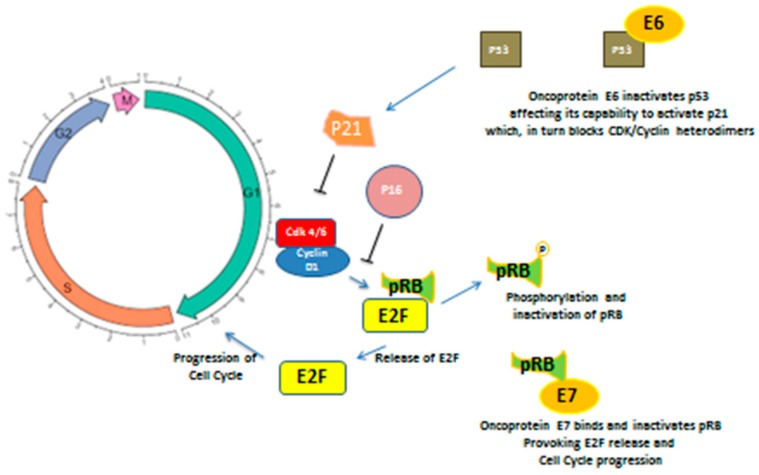
Oncoproteins E6 and E7 induce cell cycle progression acting upon cell cycle regulation, in particular during G1/S transition. E6 binds and inactivates P53, affecting its capability to activate P21, which in turn is not able to arrest CyclinD1/Cdk4/6 heterodimer. Oncoprotein E7 directly acts on RB, linking and inactivating it, leading to E2F upregulation and cell cycle progression.E2F: elongation factor 2; pRB: retinoblastoma protein; Cdk4/6: cyclin dependent kinase; p21: protein 21; E6: oncoprotein 6; E7: oncoprotein 7.

**Figure 2 cancers-12-00975-f002:**
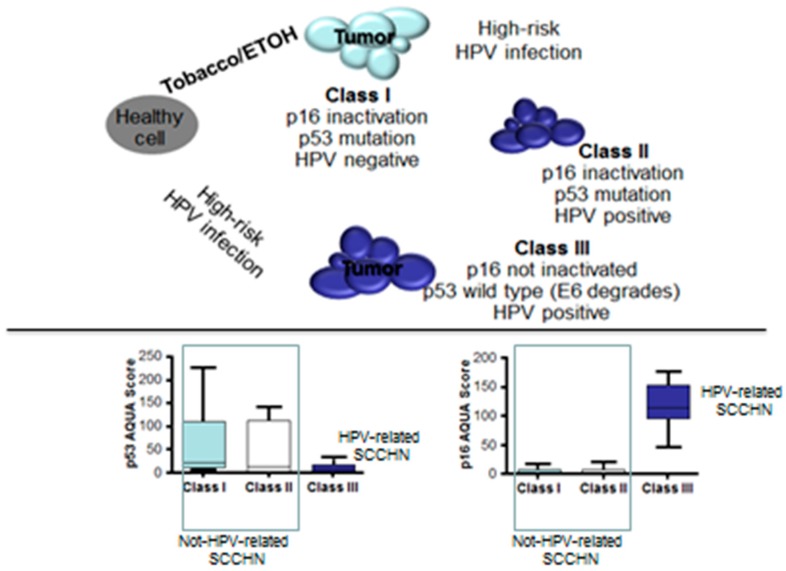
Proposed model for head and neck squamous cell carcinomas (HNSCC) carcinogenesis. Not all the human papillomavirus (HPV)-positive HNSCCs are also HPV-related. Both HPV-negative and HPV-positive but not HPV-related HSNCC (Class I and II, respectively) are characterized by INK-4 (the gene encoding for P16) and TP53 mutations which allow for a high expression of P53 and a down-regulation of P16 on the immunohistochemical assay. On the other hand, HPV-related HSNCC (Class III) show reverse features. 33. Source: Weinberger PM et al., J Clin Oncol. **2006**, *24*, 736–747 [33]. ETOH: alcohol; SCCHN: squamous cell carcinoma of the head and neck.

**Figure 3 cancers-12-00975-f003:**
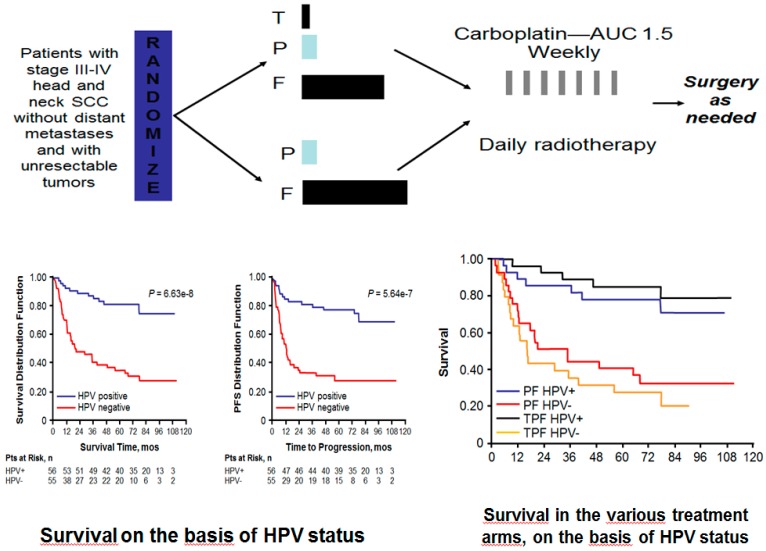
The subgroup analysis of Tax342 study revealed that HPV-positive patients treated with both induction TPF and induction PF performed better when compared with those who were HPV-negative, suggesting that HPV positivity could be considered a factor predictive of good response to sequential chemoradiotherapy. Source: Fakhry C et al. J. Natl. Cancer Inst., **2008**, *100*, 261–269 [61]. PF: cisplatin- 5Fluorouracil induction chemotherapy; TPF: docetaxel-cisplatin-5Fluorouracil induction chemotherapy.

**Figure 4 cancers-12-00975-f004:**
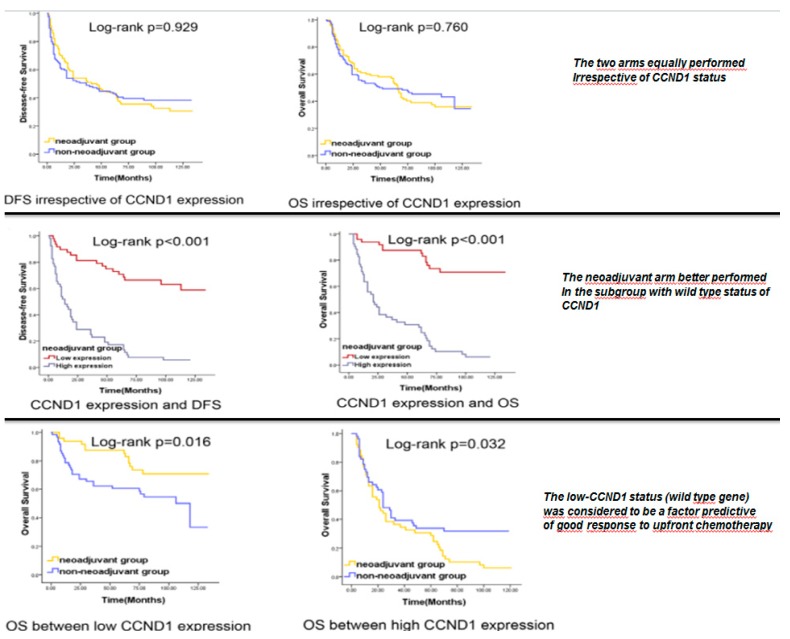
The wild type status for CCND1 is sharply linked to the HPV positivity in HSNCC. It is strongly related to the good response to sequential chemoradiotherapy and with the poor response to the upfront surgery. Source: Feng Z et al. PLoS One. **2011**, *6*, e26399 [62]. CCND1: cyclin D1; DFS: disease free survival; OS: overall survival.

**Table 1 cancers-12-00975-t001:** The picture describes the main genetic features characterizing the virus-related, the mutagens-related HSNCC and the HPV positive but not related HSNCC.

HPV-related HSNCC	Alcohol and Tobacco related HSNCC	HPV-positive (but not related) HSNCC
-P16 upregulation (not mutated INK-4 gene) -CCND1 wild type -TP53 wild type -Low number of genic/chromosomal abnormalities -Higher rate of PI3Kca mutations -Extensive TSG promoters methylation -High immune infiltrate	-P16 downregulation (INK-4 mutations) -TP53 mutations -CCND1 amplification -High number of genic/chromosomal abnormalities -Low immune infiltrate	-P16 upregulation (not mutated INK-4 gene) -TP53 mutations -CCND1 amplification -High number of genic/chromosomal abnormalities -Low immune infiltrate

CCND1: Cyclin D1; PI3Kca: the gene encoding for protein 3 Kinase; TSG: tumor suppressor genes; INK-4: INhibitors of CDK4; HPV: human papillomavirus; HSNCC: Head and neck squamous cell carcinoma.

**Table 2 cancers-12-00975-t002:** Trials exploring the concept of de-intensification of the standard chemoradiotherapy regimen in patients with HPV-positive HSNCC.

Study	Design	Type of Study	Number of Patients	Setting	Results
*Eur. J. Cancer***2020**, *124*, 178–185 (Update De-Escalate trial) [40]	cDDP-RT vs. Cet-RT in HPV-positive oropharyngeal Carcinomas	Phase III randomized trial	334	Stage II-IV oropharyngeal carcinoma	cDDP-RT better than Cet-RT in terms of 2-year OS
*Lancet***2019**, *393*, 51–60 [39]	cDDP-RT vs. Cet-RT in HPV-positive oropharyngeal Carcinomas	Phase III randomized trial	334	Stage II-IV oropharyngeal carcinoma	cDDP-RT better than Cet-RT in terms of 2-year OS and ORR
*Lancet***2019**, *393*, 40–50 [38]	cDDP-RT vs. Cet-RT in HPV-positive oropharyngeal Carcinomas	Phase III randomized trial	849	Stage II-IV oropharyngeal carcinoma	cDDP-RT better than Cet-RT in terms of OS and PFS
*Rep. Pract. Oncol. Radiother.***2018**, *23*, 451–457 [37]	cDDP-RT vs. Cet-RT in HPV-positive oropharyngeal Carcinomas	Retrospective Study	291	Stage I-IV oropharyngeal carcinoma	cDDP-RT better than Cet-RT in terms of ORR and CSS
*Oral Oncol.***2016**, *53*, 91–96 [36]	Reduced RT dose (54 vs. 70 Gy upon nodes) plus cisplatin in HPV-positive oropharyngeal Carcinomas	Phase II prospective trial	50	Stage II-IV oropharyngeal carcinoma	5-year LCR, DFS and OS were 96%, 81% and 86%
*Cancer***2018**, *124*, 2347–2354. [35]	Reduced RT dose (60 vs. 70 Gy) plus weekly cisplatin in HPV-positive oropharyngeal Carcinomas	Phase II prospective trial	44	Stage II-IV oropharyngeal carcinoma	3-year LCR, CSS, DMFS and OS were 100%, 100%, 100% and 95%
*J. Clin. Oncol.***2014**, *32*, 5s, (suppl; abstr LBA6006) [34]	Induction Cddp-Pac and Cet followed by Cet + underpowered IMRT (54 Gy) in patients obtaining a CR or a PR	Phase II prospective trial	90	Stage II-IV oropharyngeal carcinoma	70% of CR after IC 2-year PFS and OS were 80 and 94%

cDDP: cisplatin; RT: radiotherapy; IMRT: intensity modulated radiation therapy; Cet: cetuximab; Pac: paclitaxel; OS: overall survival; PFS: progression-free survival; ORR: overall response rate; LCR: locoregional failure rate; DFS: disease-free survival; CSS: cause-specific survival; DMFS: distant-metastases-free survival; CR: complete response; IC: induction chemotherapy.

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
