# Peer review of "Management of HPV-Related Squamous Cell Carcinoma of the Head and Neck: Pitfalls and Caveat"

_cancers, 2020, doi:10.3390/cancers12040975_

Round 1
Reviewer 1 Report
The manuscript is well written and interesting to read. However, there are many minor details which should be given attention before publication.
- Often blanks are missing, see lines e. g. 33, 38, 66. Please check entire MS.
- Inconsistant citation, e.g. line 142: Chena BS et al., or 150: Woody NH et al. Please check entire MS for this.
- In line 109 you cirte Weinberger et al but only at the end of para you bring the right number. This should always be done in the sentence you cite the author for the first iem and not at the end of the para. Several times. Please checke entire MA.
- line 85: damage. (25, 26) cghange to damage (25,26). Several times in ms, please check
- No reference to Fig. 1 in text. Legend to Fig. 1 needs more explanation in order to be understandable.
Author Response
Dear reviewer,
we have willingly accepted your suggestions and we have provided to resolve the questions raised from you, in particular:
- we have performed a revision of the entire manuscript, correcting the mistakes
- we have modified the bibliography according to your suggestions
- we have corrected the reference number 33 (and all the similar), adding it in the position indicated by you
- we have corrected the paragraph at the line 85, as you have requested
- we have inserted in the text the sentence recalling the Figure 1 (which following the revisions, has been renamed "figure 2), and moreover we have modified the Figure legend as you have requested
Best regards
Dr Francesco Perri (Corresponding author)
Reviewer 2 Report
Re: Manuscript cancers-758854
Comments to the authors:
This is an interesting manuscript that reviews some of the literature that compares HPV positive and HPV negative head and neck squamous cell carcinomas. The authors provide some level of detail about the genetics of HPV positive/related tumours but with very little comparison to HPV negative cancers. The review article also provides some information about clinical trials that used reduced radiotherapy doses for HPV positive tumours. While the review discussed some aspects of the future of therapy, there was very little about Immunotherapy. The focus of the review article (according to the title) is the management, and therefore immunotherapy should be covered in more detail.
Although the topic of the review article is very interesting and is of great importance from a clinical point of view, the paper is not very well written and requires extensive revision. Some of the sections are rather superficial and lack depth. Given the wealth of literature available about HPV positive cervical cancer, it is surprising that the authors did not include comparisons to oropharyngeal carcinomas.
The following are the specific comments:
- In the background section (and the abstract), the authors focus on alcohol, tobacco and trauma as the main etiological factors for HPV negative head and neck cancer. Given that the highest incidence globally is in countries that use betel quid, this needs to be highlighted as a risk factor and perhaps even a different disease entity.
- Also, in the background, the authors say that HPV positive head and neck caner have a male predominance as a unique feature, however, that is a similar trend in HPV negative cases as well, albeit with different male: female ratio
- There are many typographical, linguistic and grammatical errors throughout the manuscript that need to be addressed.
- What is the source of data in the lower panel of Figure 1?
- Figure 1 caption does not describe the bottom panel in the figure (the bar graphs with the p53 and p16 scores). Please provide information about this.
- In section2, the genetics of HPV positive HNSCC, it would be really helpful to add a figure that describes the pathways involved in the pathogenesis.
- IN the same section, there is practically no comparison between HPV positive/related and HPV negative tumours, which is the aim of the article. The authors offer some explanation as to the possible difference between HPV related and HPV positive tumours, but very little comparison to the HPV negative tumours.
- Figure 2, please cite the source of the data presented.
- Figure 3, please cite the source of the data.
- Figure captions need to be reviewed
- Table 1 practically repeats the same information in section 3
- The section about immunotherapy is lacking. There are many studies that looked at immunotherapy in head and neck cancer, so these should be discussed as they represent a very important development in cancer therapy.
- Section 5 should be placed before section 4
Author Response
Dear reviewer,
we have accepted with interest the corrections and the suggestions that you have provided to us. We have performed all the requested corrections and in particular:
1. We have added in the background section also a statement highlighting the "betel quid chewing" as a risk factor for the developement of HSNCC
2. in the background section, we have corrected our statement, highlighting that also HPV negative counterpart shows an high male/female ratio
3. we have performed a wide revision of the english grammar, requiring also the help of a collegue with english-editing expertise
4. we have added the source data in the Figure 1 (which has been re-named Figure 2, following the revisions) legend
5. We have better explained, in the Figure legend, the significance of the Figure 1 (re-named Figure 2)
6. We have added a figure (Figure 1) describing the main genetic pathways involved in HPV-related and HPV-negative HSNCC
7. We have added in the 2 chapter "genetics of HPV positice HSNCC" a section regarding the genetics and the common features of the HPV negative HSNCC, as you requested
8. We have added the source of the Figure 2 (which has been changed in Figure 3, after the revisions)
9. We have added the source of the figure 3 (which has been changed in Figure 4, after the revisions)
10. We have revised the figure captions
11. We have decided to leave the Table 1, with the aim to summarize the concepts graphically and quickly
12. We have added two section explaining better the impact of immunotherapy, both in the chapter 4 and in the chapter 5 (highlighting in the last, the clinical data present in the literature)
13 we have posed chapter 5 in the site of the 4, as you have requested
Best regards
Dr. Francesco Perri (corresponding author)
Reviewer 3 Report
In this manuscript, the authors reviewed and summarized the genetics of both the HPV-positive and the HPV negative HNSCC, discussed its impact on the clinical behavior and on the therapeutic strategies. It was well written and helpful. The only comment here is that the authors should talk more about the differences of the molecular alterations between the HPV+ and HPV- HNSCC.
Author Response
Dear reviewer,
we have accepted with enthusiasm your suggestions and we have tried to modify the manuscript according to them. In particular:
we have better explained in the text, in particular in chapter 2 "genetics of HPV positive HSNCC" the sharp difference between HPV positive and HPV negative tumors on the basis of their genetics
Best regards
Dr. Francesco Perri (corresponding author)
Round 2
Reviewer 2 Report
Re: Manuscript cancers-758854-V2
Comments to the authors:
The authors have improved the manuscript and responded to many of the comments satisfactorily. However, the paper remains poorly written and the language requires significant editing.
There are few points that require attention.
- The Authors’ response to my previous comment 6, about the usefulness of including a figure that describes the pathways involved, is not satisfactory. They provided a table rather than a figure and called it Figure 1, without describing actual pathways. My suggestions were for a figure that describes the actual pathway (How E6 inactivates p53 and then p21 and the effect on CDK/Cyclin and then the effect on pRB and E2F…..etc).
- In relation to the current Figure 1, it is good to keep it as a table to describe the genetic differences between HPV negative and HPV positive tumors, adding to that the differences between HPV related and HPV positive but not related, so that you have three columns.
- The authors did not provide any real comparison to HPV related cervical cancer.
- The manuscript is poorly written and there are still many typographical errors throughout the manuscript.
- This is a minor formatting point, but the figures have a lot of red lines indicating errors from the software used to generate the files. These can be easily removed.
Author Response
Dear reviewer, we appreciated your further suggestions and we have modified our manuscript according to them, in particular:
- The Authors’ response to my previous comment 6, about the usefulness of including a figure that describes the pathways involved is not satisfactory. They provided a table rather than a figure and called it Figure 1, without describing actual pathways. My suggestions was for a figure that describes the actual pathway (How E6 inactivates p53 and then p21 and the effect on CDK/Cyclin and then the effect on pRB and E2F…..etc).
- Response: We have added the requested figure (named figure 4)
- In relation to the current Figure 1, it is good to keep it as a table to describe the genetic differences between HPV negative and HPV positive tumours, adding to that the differences between HPV related and HPV positive but not related, so that you have three columns.
- Response: We have added the figure showing the differences between HPV-related, HPV-not related and HPV-negative tumors (named Table 1)
- The authors did not provide any real comparison to HPV related cervical cancer
- We have voluntarily decided to not describe cervical cancers because the hot topic regards only head and neck carcinomas. We apologize for the mistake.
- The manuscript is poorly written and there are still many typographical errors throughout the manuscript.
- We have further revised the manuscript, giving it to another expert
- This is a minor formatting point, but the figures have a lot of red lines indicating errors from the software used to generate the files. These can be easily removed.
- We have tried to delete this error
Best Regards and thanks a lot for your suggestions
Dr. Francesco Perri (Corresponding Author)
Round 3
Reviewer 2 Report
Re: Manuscript cancers-758854-V3
Comments to the authors:
The authors have improved the manuscript and responded to the comments satisfactorily. The manuscript reads much better than before.
There are still some formatting and typographical/linguistic errors that can be edited.